# Carrier-envelope phase on-chip scanner and control of laser beams

Václav Hanus [1] ✉, Beatrix Fehér [1], Viktória Csajbók [1], Péter Sándor [1], Zsuzsanna Pápa [1,2], Judit Budai[2], Zilong Wang[3,4], Pallabi Paul [5,6], Adriana Szeghalmi [5,6] & Péter Dombi [1,2] ✉

The carrier-envelope phase (CEP) is an important property of few-cycle laser pulses, allowing for light field control of electronic processes during laser-matter interactions. Thus, the measurement and control of CEP is essential for applications of few-cycle lasers. Currently, there is no robust method for measuring the non-trivial spatial CEP distribution of few-cycle laser pulses. Here, we demonstrate a compact on-chip, ambient-air, CEP scanning probe with $0.1\ \mu m^3$ resolution based on optical driving of CEP-sensitive ultrafast currents in a metal−dielectric heterostructure. We successfully apply the probe to obtain a 3D map of spatial changes of CEP in the vicinity of an oscillator beam focus with pulses as weak as 1 nJ. We also demonstrate CEP control in the focal volume with a spatial light modulator so that arbitrary spatial CEP sculpting could be realized.

The evolution of the electric field of ultrashort laser pulses plays a crucial role in the dynamics of laser-matter interactions. Contemporary laser technology allows for straightforward control of the field shape through a property called the carrier-envelope phase (CEP), which describes the phase of the carrier wave with respect to the intensity envelope of a laser pulse. Robust techniques such as f-to-2f interferometry[1–4] and above-threshold ionization stereometry[5] enable laser users to CEP-stabilize laser pulse trains[2,3] or measure CEP in phase tagging experiments[6–9]. This has paved the way for the observation of many intriguing light-phase dependent phenomena related to photoionization of atoms[10,11] and molecules;[11,12] photoemission from metals[13,14] and metallic nanoparticles;[15,16] high-harmonic generation[17,18], photochemical reaction directional control[19,20], and optically induced current[21–23]. At the same time, the ability to directly access the evolution of the electric field shape through CEP has been a key ingredient in attosecond science[24]. However, as most experiments are conducted in the vicinity of the laser focus where the spatial distribution of CEP is non-trivial[25,26], demonstrations of CEP-sensitive interactions with larger structures, such as photoemission from arrays of nanoparticles[15,27–29], surface-enhanced Raman scattering on adsorbed molecules[30], high-

harmonic generation from gases and especially solids[31,32], current in reconfigurable circuits[33], or petahertz networks[23,34], are still pending, as the scaling of such systems is limited due to volume smearing effects. Here, we present a method for measuring and controlling CEP spatial distributions that enable the establishment of areas of well-defined CEP volumes. This will allow for the study of CEP effects in larger systems, which could become promising cornerstones for the development of attoscience-based real-life devices.

Some successful measurements of CEP spatial distributions have already been achieved by scanning a CEP-sensitive probe within the focal volume of a laser beam. One approach exploits the CEP sensitivity of a bunch of electrons emitted from a nanotip in vacuum[25]. Another approach based on interferometry[35] relies on a superposition of sample and reference beams as a function of their respective delay. However, these methods have not found much utilization due to limitations, such as the need for a vacuum apparatus (nanotip method) or an interferometric setup. This calls for the design and implementation of compact, and easy-to-operate CEP devices for single-beam, non-interferometric and real-time measurements.

[1]Wigner Research Centre for Physics, 1121 Budapest, Hungary. [2]ELI-ALPS Research Institute, 6728 Szeged, Hungary. [3]Physics Department, Ludwig-Maximilians-Universität, 85748 Munich, Germany. [4]Max Planck Institute of Quantum Optics, 85748 Garching, Germany. [5]Institute of Applied Physics, Abbe Center of Photonics, 07745 Jena, Germany. [6]Fraunhofer Institute for Applied Optics and Precision Engineering, 07745 Jena, Germany. ✉e-mail: hanus.vaclav@wigner.hu; dombi.peter@wigner.hu

In recent years, several CEP-sensitive on-chip devices were demonstrated, which have the potential to serve as the basis of an ambient-air and non-interferometric probe thanks to their small size. One such device is based on nanotunneling of electrons from a nanoparticle over a nm-sized junction to an anode[36,37]. An array of nanoparticles has already been proven as a viable option to detect the CEP of laser beams[38,39], but no spatial resolution was demonstrated. Another method is based on the generation of light-field-driven ultrafast currents in dielectrics or semiconductors[22,40]. The nonlinear strong-field interaction between the laser and electrons in the medium couples the valence and the conduction band, rendering the medium virtually conductive[23,41] for the duration of the strong laser field. Placing metal electrodes in the vicinity of the interaction volume enables the detection of currents whose magnitude and direction are CEP-dependent. A solid-state CEP measurement device was proposed based on this phenomenon[7].

Here, we exploit the capability of light-field-driven ultrafast currents in solids[23] to measure spatial changes of the CEP and we construct an on-chip, ambient-air, single-beam, and compact probe with sub-micron resolution. Our new probe significantly differs from our earlier design published in ref. 23 so that a CEP change scanner device can be constructed. We implemented tapered electrodes, see Fig. 1a, b, on a thin metal–dielectric heterostructure of Ir and $Al_2O_3$ made by atomic-layer deposition (ALD) on a fused silica substrate[42]. The heterostructure has a nanolaminate structure and possesses a high $\chi^{(3)}$ that enhances the magnitude of CEP-sensitive currents, as already indicated in refs. 23,43. (Note that the electrodes are far enough from each other so that no electron tunneling can take place between them unlike in ref. 44). This way, we are able to characterize CEP distributions of focused laser beams formed by pulses having energy as low as 1 nJ. In combination with either spectral phase shaping or wavefront shaping using a spatial light modulator (SLM, Hamamatsu X15213-07), we measured and controlled CEP distributions in the vicinity of the laser beam focus.

## Results

### On-chip, ambient-air CEP probe

The design of the microscopic, ambient-air scanning CEP probe needs to satisfy two aspects: first, the active volume that is sensitive to the CEP needs to be small enough to allow resolution of the laser beam; second, the proportionality of the acquired phase being a function of CEP needs to be constant over a large range of intensities. The former was achieved by making lithographic gold electrodes in a shape of two opposing tapers and we estimate the volume resolution of our probe to be $0.1\,\mu m^3$ and a lateral resolution of $0.6\,\mu m$ (determined by the electrode distance), see Fig. 1b and "Methods" for details on geometry. When illuminated with a laser beam phase-locked to a nonzero $f_{CEO}$ frequency (1 kHz), our on-chip CEP probe generates an alternating current, whose amplitude $J_O$ and phase $\varphi_J$ can be measured with a lock-in amplifier referenced to $f_{CEO}$. The current is steady exhibiting only $3.5°$ of standard deviation in $\varphi_J$ in the time interval of 1 min, see Fig. 2d.

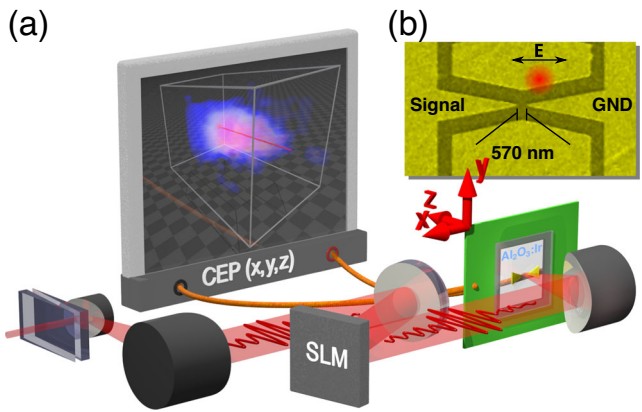

**Fig. 1 | Scheme of the experimental setup. a** The laser beam propagates through dispersion-control wedges and bounces off mirrors and the SLM. The focus of the beam after the parabolic mirror is scanned by XYZ positioning of the phase-scanner probe in the focal volume and the CEP($x,y,z$) function is acquired (see 3D colour map as a cloud on the screen). The probe is connected via a transimpedance amplifier and a lock-in amplifier, so that the current $J_O$ and its phase $\varphi_J$ could be measured at a reference frequency $f_{CEO}$. $\varphi_J$ is then a direct measure of the CEP. **b** The SEM image of the design of the microscopic probe: Two gold electrodes denoted as *Signal* and *GND* shaped to a tapered ending are separated from each other by a ~570 nm gap. The red circle illustrates the impinging laser beam, and it is depicted in-scale with respect to the SEM image (the edge of the circle is at $e^{-2}$ of the typical intensity profile). The black arrow depicts the direction of the laser polarization.

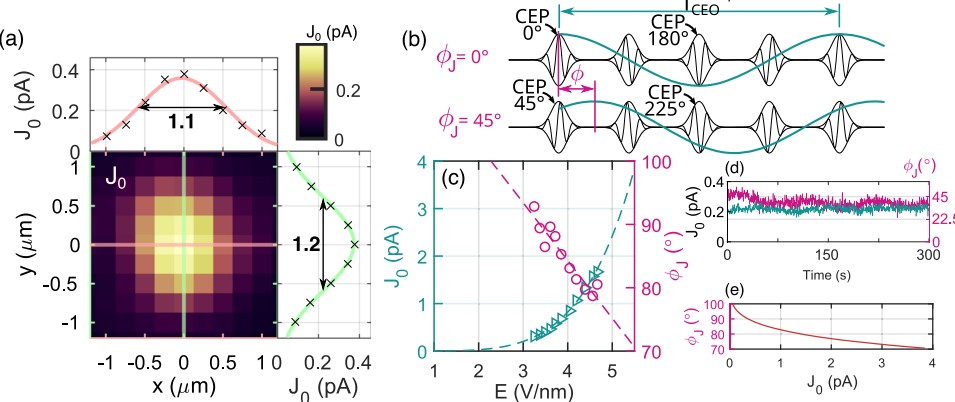

**Fig. 2 | Current and phase response of the on-chip probe. a** The heatmap shows a typical current magnitude $J_O$ profile resulting from an $xy$ position scan in the laser polarization plane at the depth where the intensity is the highest. Side plots show cut-outs through the $J_O$ distribution vertically (green) and horizontally (pink) with respective Gaussian fits and the $1/\sqrt{e}$ diameter. **b** Illustration of measured lock-in phase $\varphi_J$ and CEP equivalence. The two depicted pulse trains differ in CEP offset by 45°. This projects to the offset of the current oscillation in the connected circuit (green line) leading to the change in the detected lock-in phase $\varphi_J$ by 45°. **c** Measured lock-in current magnitude $J_O$ (green triangles) and phase $\varphi_J$ (pink circles) as they are generated during the illumination of the CEP probe. The CEP probe and beam are kept stationary, while the power of the beam is changed resulting in the change of the electric field amplitude $E$. Dashed lines represent power law and linear fits, respectively. **d** Temporal stability of the current $J_O$ (green) and the phase $\varphi_J$ (pink) from the probe tested for $t = 300\,s$ with 3 Hz acquisition rate. **e** The relationship between the fits of the measured current and the phase $\varphi_J(J_O)$ which is used as a correction to the measured CEP scans.

As concluded in our previous work[23], $\varphi_J$ follows the relative CEP offset of the laser pulse train. Hence, we label the measured phase $\varphi_J$ as $\Delta$CEP in the following figures. For better understanding of the validity of this equivalence, we provide an illustration in Fig. 2b, where two pulse trains on the picture represent two different spatial points with respective CEP offset of 45°. Consequently, the measured $\varphi_J$ difference between these two points will be 45° as well. It turns out that $\varphi_J$ has a slight dependence on the laser electric field: 7.5 °nm/V, see Fig. 2c. We compensated for this property by introducing an empirical $J_O$ dependent offset $\varphi_J(J_O)$, see Fig. 2e and Supplementary information for details.

In order to demonstrate the sensitivity of our nJ-level CEP probe, we investigated a focus of a 2.5 nJ broad-spectrum oscillator laser beam providing pulses with $\tau = 5.5$ fs duration (intensity full-width at half-maximum, FWHM) at a repetition rate of 80 MHz, see Fig. 1a for a sketch of the setup. Due to losses in the beamline the typical pulse energy during the experiment was about 1.5 nJ. After a tight focusing (see "Methods") we expect a peak intensity of up to $13 \times 10^{12}$ W/cm², i.e., 7 V/nm field strength, and we obtain a CEP-sensitive current without any optical damage to the probe (note, that the CEP-dependent current should be observable already from 0.6 V/nm as indicated in ref. 23). A scan of $J_O$ with 250-nm-long steps in the polarization plane in the focal position (defined as a point on the optical axis where $J_O$ is the highest) reveals the resolution and range of the CEP probe in the polarization plane, see Fig. 2a. A measurable signal was acquired in the range $(-1.0w_O, +1.0w_O)$, $w_O = 1.5$ μm, with a useful scanned area of 6.9 μm². Note that the polarization is linear, and the electric field vector is parallel to the line connecting the two opposing electrodes. If the polarization is set perpendicular, the current drops almost to zero, see Supplementary information.

## Oscillator beam spatial CEP distribution

The spatial $\Delta$CEP$(x, y, z)$ distribution of a short, broad-spectrum beam can acquire various shapes as measured in earlier studies[25,26,35], and supported by theoretical studies[26,45–47]. In Fig. 3, we show a complete volumetric characterization of the CEP distribution around the laser beam focus ($w_O = 1.7$ μm) as we scanned a useful volume of 84 μm³. Here, we use the following convention for the spatial coordinates: $x-$horizontal (laser polarization direction), $y-$vertical, $z-$optical axis, see Fig. 1a. To visualize the measurement, we have cut the measured 3D distribution into three slices intersecting at the point of highest current $J_O$ which is assumed to be the focus of the beam. We can see that in the laser focal plane, see Fig. 3c, which is also a plane of the highest laser intensity, the $\Delta$CEP is not constant and tends to decrease in value outwards. In planes along the optical axis, see Fig. 3a, d, the CEP forms a ridge along the optical axis $z$, and the phase drops along $z$. The drop of CEP is asymmetric with respect to the point of highest $J_O$ ($z = 0$ μm) and ranges from 30° to −80° (extracted in Fig. 3c). This contrasts to a purely Gouy phase shift, which shows a symmetric drop of the CEP value.

To explain our observations, we need to consider chromatic aberrations that have a crucial influence on the CEP distribution[45,46]. These aberrations can be approximated by introducing parameters $g$, $\gamma$, and $C_r$ (chirp), which correspond to the first derivatives with respect to the frequency evaluated at the central frequency of: beam waist, focal position, and spectral phase, respectively, see "Methods" for their definition. The assumption of a Gaussian beam TEM$_{00}$ leads to an analytical model derived in ref. 45 and reproduced in Methods, Eq. (2). We compared the measurement with the model evaluated for certain $g$, $\gamma$, and $C_r$ values. We found the best agreement for values: $g = 0.2$, $\gamma = 0.3$ and $C_r = -0.8$, see Fig. 3b, e, f. Despite performing the scan at a pulse compression setting where the CEP-sensitive signal $J_O$ reached its peak and maximum pulse compression was expected, it is noteworthy that a negative chirp value in the model was still necessary to match the measurements. This discrepancy can be explained by the negative

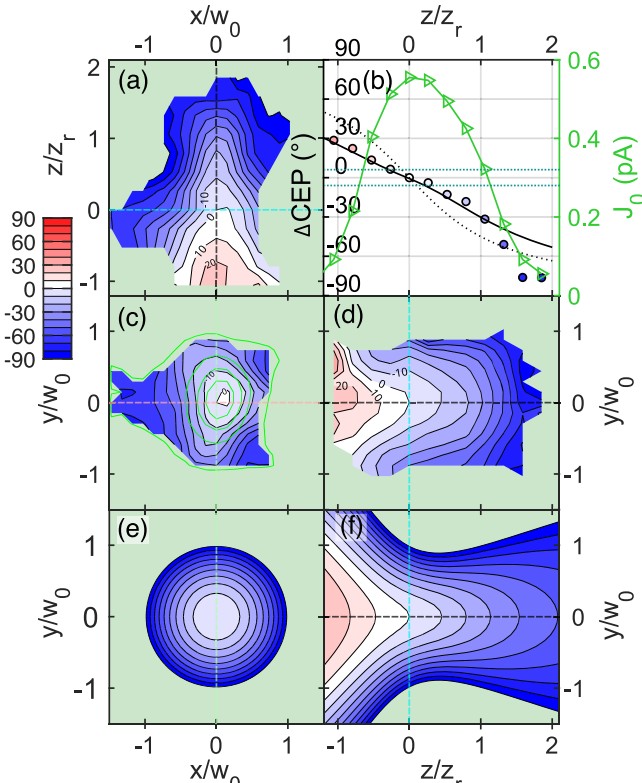

**Fig. 3 | Spatial CEP distribution measured with the on-chip probe. a**, **c**, **d** Cross-sections through the measured spatial change of CEP in degrees in the *xz* (horizontal), *xy* (transversal), *yz* (vertical) planes, where *x* and *y* are the axes along (horizontal) and perpendicular (vertical) to the laser polarization, respectively, and *z* is the laser beam propagation axis. The beam waist and the Rayleigh length of the measured beam are $w_O = 1.7$ μm and $z_r = 11.3$ μm, respectively, with a central wavelength of 800 nm. **c** The phase map is overlapped with green contours of the measured current $J_O$. Cuts go through the point of the highest $J_O$. Dashed lines show the intersects of the plotted cuts: cyan for transversal, pink for horizontal and green for vertical. Phase and $J_O$ along the optical axis (black dashed line) are shown in (**b**). Triangles and circles are the measured datapoints of the current amplitude $J_O$ and the CEP, respectively. The maximum of $J_O$ (green) signifies the position of the focus, where the measured phase was offset to zero. Solid black line shows the best fit of the model from Eq. (12) in ref. 45 with parameters $g = 0.2$, $\gamma = 0.3$, and $C_r = -0.8$. Dotted line is the $-\operatorname{atan}(\frac{z}{z_r})$ function, i.e., Gouy phase. Dotted horizontal lines define the range of CEP used for the calculation of volume of flat phase being 9.6 μm³. **e**, **f** Show the model in *xy* (transversal), *yz* (vertical) plane.

curvature of the spectral phase at the central frequency, as indicated by the spectral phase reconstruction based on the dscan measurement. The negative value of $C_r$, corresponding to this negative chirp, can be attributed to the inherent challenge of achieving an ideally flat spectral phase for an octave-spanning laser pulse. Additional details can be found in the Supplementary information.

## Arbitrary CEP control in the focal volume

One viable scheme to shaping of CEP landscape is the spectral phase manipulation accessible via pulse chirping[45]. The theoretical investigations suggest that the presence of a non-flat spectral phase can lead to a significant bending of the phase profile in both axial and transversal direction. Indeed, the ridge structure presented in Fig. 3a, c, d was reproduced with a negative $C_r$ value in the model (Fig. 3e, f). To illustrate the capabilities of the chirp to control the $\Delta$CEP$(x,y,z)$, we performed a measurement of CEP landscape for two positions of dispersion compensation wedges. The first settings, denoted as $d = 0$ mm, produced maximum current, while the other setting was produced by adding 0.5 mm fused silica glass in the beamline, i.e., $d = 0.5$ mm. This

corresponds to adding 18 fs² of group delay dispersion to the spectral phase $\varphi(\omega)$ and relative chirp $C_r$ change of 0.8. The range of chirping is limited as with more chirp the laser pulse becomes longer, and consequently the signal from the probe becomes weaker, due to an increasing symmetry of the electric field under the envelope of the pulse. Results are presented in Fig. 4. Looking at the measured CEP distribution in Fig. 4, we note remarkable spatial changes that resulted from the addition of positive group delay dispersion: First, in the axial CEP distribution we can see an increase of steepness of the phase in the prefocal region and decrease of steepness after the focus, see Fig. 4c. Second, more glass resulted in flattening of the ridge structure observable in the horizontal $xz$ slice, compare Fig. 4a2, b2. Evaluation of the model with $C_r = -0.8$ and $C_r = 0.0$ (with parameters $g = 0.4$, $\gamma = 0.1$) captures well the qualitative change of shape between cases $d = 0$ mm and $d = 0.5$ mm, see Fig. 4a3, b3. As a consequence of the ridge reduction the area of flat CEP (CEP in the interval (0°, 10°)) in the vicinity of the intensity maximum increased. This flattening is very well

visible on the transverse slices in polarization plane $xy$, compare Fig. 4a1, b1.

Another way toward CEP landscape shaping is to modify the chromatic aberrations of the laser beam, which can influence $g$ and $\gamma$. This can be achieved with lenses[45], but it is not feasible with few-cycle pulses. Instead, we exploited the chromatic behavior of the liquid-crystal on a silicon spatial light modulator (SLM) placed before the final focusing element in our beamline, see Fig. 1a. The SLM device introduces a phase shift as a function of position in the near-field and this phase shift is frequency (wavelength)-dependent. If a curved pattern is set on the SLM, it imprints a curvature on wavefronts and, in turn, causes a weak lensing effect on the laser beam. The curvature is frequency dependent, and this way the size of the beam $W_O(\omega)$ on the final focusing parabola is changed. Consequently, focal waists $w_O(\omega)$ and positions of foci $f(\omega)$ change, which are linked to $g$ and $\gamma$. This can be particularly useful for laser beams produced by hollow-core fiber compressors that can carry a certain amount of chromatic aberration[48]. The frequency dependent wavefront curvature induced by an SLM can be enough to invert the dependence of wavefront curvatures on wavelength, see the Supplementary information, for a discussion on SLM limitations.

To test the $\Delta$CEP landscape manipulation with SLM, we applied the following curved pattern on the SLM as depicted in the inset of Fig. 5b. From the 3D scan of the focal volume with the CEP probe, we can see that the $\Delta$CEP landscape indeed changed, as evidenced by a

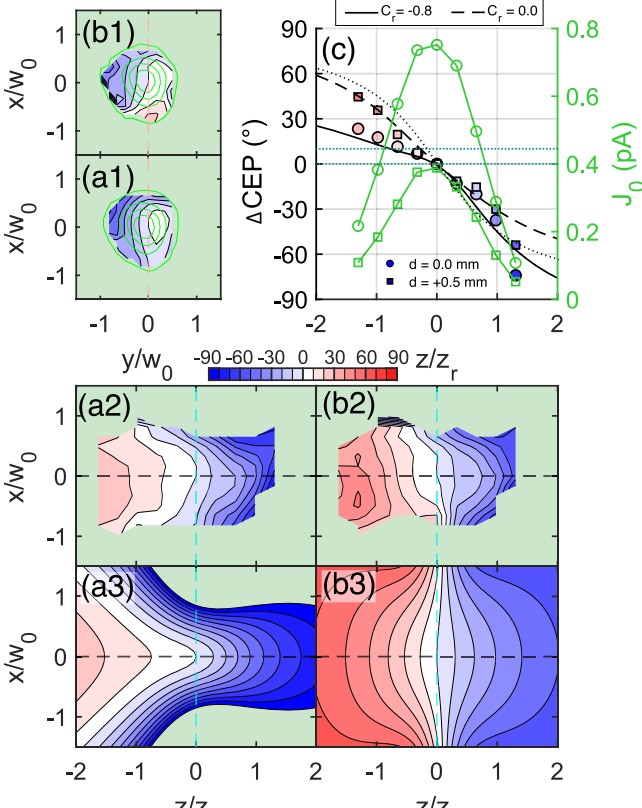

**Fig. 4 | Effect of pulse chirp on the CEP landscape.** Measured spatial change of the CEP in degrees in $xz$ plane, where $x, y$ are in-polarization horizontal and vertical axis, respectively, and $z$ is the laser beam propagation axis. The beam waist and the Rayleigh length of the measured beam was $w_0 = 1.5$ μm and $z_r = 8.8$ μm, respectively, with a central wavelength of 800 nm. Phase maps (**a1, a2**) were acquired for a pulse compression setting that provided the highest $J_O$ current, case denoted as $d = 0.0$ mm. Phase maps (**b1, b2**) were acquired for a pulse compression created from (**a**) by adding 0.5 mm fused silica in the beamline, this case is denoted as $d = 0.5$ mm. **a3, b3** show the model, evaluated for parameters $C_r = -0.8$ and 0.0, respectively. Other parameters in the model are the same and acquire values: $g = 0.4$, $\gamma = 0.1$. **c** We show the phase along the optical axis, which is highlighted in (**a2**) and (**b2**) as a black dashed line−circles: $d = 0.0$ mm and squares: $d = 0.5$ mm case. We accompany the datapoints with modeled phase values evaluated for chirp $C_r = -0.8$ and 0.0, in solid and dashed line, respectively. The dotted line is the Gouy phase for a reference. Green lines are the $J_O$ current profiles of $d = 0.0$ mm (cirles) and $d = 0.5$ mm (squares) case. The solid line is at the same time a lineout of the calculated CEP map in (**a3**) and (**b3**). Dotted horizontal lines in (**c**) define the range of CEP used for the calculation of the volume of flat phase.

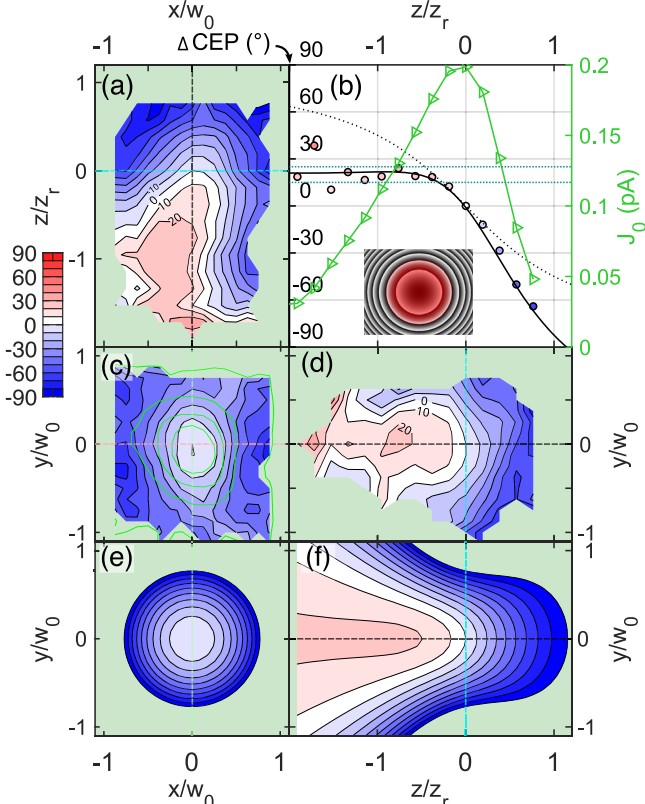

**Fig. 5 | Exploiting chromatic aberrations to flatten the CEP.** CEP phase flattening resulting from the beam shaping by the introduction of a chromatic aberration. Same representations as in Fig. 3. The beam waist and the Rayleigh length of the measured beam are $w_O = 2.0$ μm and $z_r = 15.7$ μm, respectively. The measured CEP map shows clearly a phase flattening in the prefocal region (**a–d**). The lineout along the black dashed line is shown in (**b**) with circles. Model in (**e, f**) and represented with the solid line in (**b**) is evaluated for values $g = 0.4$, $\gamma = -0.4$ and $C_r = -0.8$. Inset in (**b**) shows the applied SLM pattern with a red spot illustrating the beam size for scale. Dotted blue horizontal lines define the range of CEP used for the calculation of the volume of flat phase yielding 18.4 μm³.

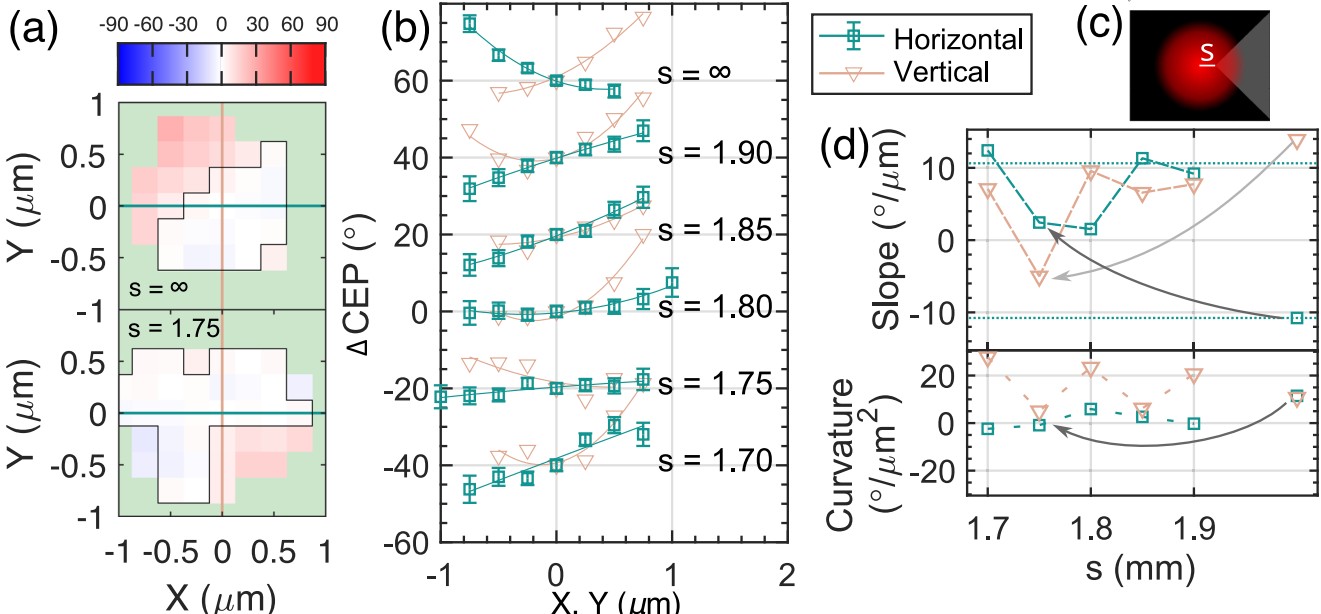

**Fig. 6 | Controlling the slope of the CEP in the polarization plane. a** Measured CEP change distribution in degrees in the plane perpendicular to the optical axis for two selected settings of SLM: $s = \infty$ (top) and $s = 1.75$ mm (bottom). The latter case illustrates how the application of SLM pattern improved the flatness of the phase along the horizontal (green line) direction. Light-green mask shades a region where signal is too low and phase measurement is obscured. Origin of the coordinate system is set to the point of largest $J_O$. Black contour encircles the region where the CEP value is in the interval (−5°, +5°). **b** CEP along horizontal (green squares) and vertical (beige triangles) slice as indicated in (**a**) for a range of values $s$. CEP is offset by 20° for every $s$ for the sake of legibility. Measured points are accompanied by polynomial fits. The length of error bars is proportional to $J_O^{-1}$. **c** Pattern applied on SLM with representation of parameter $s$. The red spot represents the size of beam used in the experiment in-scale. **d** Slope and curvature of the phase profiles from (**b**) obtained from fits for range of values $s$ from an interval (1.7, 1.9) mm. The rightmost point is the case with zero SLM pattern. Arrows highlight the improvement in the slope and curvature when switching from zero pattern to SLM pattern with $s = 1.75$ mm.

comparison of Fig. 3a, c, d with Fig. 5a, c, d. Strikingly, we identify CEP flattening effect in the prefocal region, as the axial phase keeps steady at average value of $20 \pm 5°$ in the axial range $(−1.7z_r, −0.3z_r)$, $z_r = 15.7\,\mu m$. This feature is followed by a significant drop in phase value from 0° to −60° in $(0.0z_r, 1.4z_r)$. In the prefocal region, the flattening also extends in the lateral direction, see Fig. 5a, d, leading to a continuous and convex volume of $18.4\,\mu m^3$ of flat phase settled between values (15°, 25°). This is a significant improvement of the volume of approximately constant CEP compared to the case without the wavefront manipulation: The flat phase volume of the case presented in Fig. 3 yields only $9.6\,\mu m^3$ of phase settled between values (−5°, 5°). Furthermore, we have compared results of modified beam with a model. Its evaluation with parameters: $g = 0.4$, $\gamma = −0.4$, $C_r = −0.8$ gives a good agreement with the measurement, see Fig. 5b, e, f. We note the drop of $\gamma$ by 0.7 in comparison with the measurement presented in Fig. 3 where the SLM was set to zero. This is well in agreement with the expectation that the chromatic properties defined by the size $W_O(\omega)$ of the beam on the final focusing parabola changed due to the use of the SLM.

In order to demonstrate the ΔCEP distribution sculpting using an SLM, our objective was to achieve a controlled change in the slope of the ΔCEP along the horizontal $x$ axis. A trial calculation of a Fresnel integral showed that it is possible to achieve a horizontal tilt of the focal phase. This can be accomplished by applying a phase offset of π on the wavefront of the pupil electric field in two distinct regions, which are separated by a V-shaped border, see Fig. 6c and "Methods" for the discussion on the relation between the pupil function and CEP. To test this pattern, we first measured the distribution of ΔCEP in the plane perpendicular to the laser propagation direction when no SLM pattern was applied, this case is denoted as $s = \infty$. In Fig. 6a top, we show the ΔCEP in a point along the optical axis $z$ where $J_O$ was the highest. From this data we plot the ΔCEP along the horizontal and

vertical directions through that point, see Fig. 6b. Finally, we applied a polynomial fit to acquire a measure of the slope and the curvature of the ΔCEP profile, see Fig. 6d. One can see that using a beam as is (without the SLM manipulation), we did not get a flat CEP response (see Fig. 6a top profile, and Fig. 6b topmost curve for $s = \infty$ case). Then the predefined SLM pattern was applied, and an $xy$-scan was performed in the same $z$ position as in the case without the pattern. We varied the only free parameter $s$, which represents the distance of the V-shape apex from the center of the SLM chip in millimeters. We can see in Fig. 6b that the horizontal CEP profile indeed changes with changing the SLM pattern, while the optimal setting was found for $s = 1.75$. The exclusivity of this point for our setup is confirmed also with the low value of the slope and curvature obtained from the fits. In addition, we show the full CEP profile in the scanned $xy$-plane for this $s$ value in Fig. 6a bottom. Apart from flattening the phase along a line, the area of constant phase, defined as an area where the phase is the interval between (−5°, +5°) increased from 1.06 to $1.63\,\mu m^2$ by applying the optimal SLM pattern. Our experiment thus shows that it is possible to use a feedback loop to gain control over the CEP landscape.

## Discussion

We demonstrated a compact, on-chip, and single-beam CEP scanner device operating under ambient laboratory conditions. The technique relies on the generation of CEP-sensitive ultrafast currents in a thin nanolaminate coating. The currents are detected using a microscopic electric probe with a sensitive volume of $0.1\,\mu m^3$. The probe is assembled on an optical chip. We anticipate that our technique is compatible with a wide range of laser systems providing laser pulse energy >1 nJ as long as the measured laser pulse is of few-cycle character and they are phase-locked to a nonzero $f_{CEO}$. The applicability to systems that lock to $f_{CEO} = 0$ can be ensured by, for example, introducing a piezo-driven mechanical CEP modulation with wedges.

We also showed that the use of an SLM is a viable option to sculpt the CEP distribution of a laser beam. We demonstrated a control of the spatial change of the CEP, including CEP flattening in the focal volume. We propose that the CEP probe could be a part of a feedback mechanism that would ultimately enable arbitrary CEP sculpting. We foresee that CEP scans will pave the way towards observations of new phenomena that were previously hindered by volume CEP smearing and would otherwise have to rely on the signal from small volumes.

## Methods

### Active material and electrode design
For the CEP-probe fabrication we utilized a 1D-nanolaminate-coated slab on top of which we patterned gold electrodes using electron beam lithography. The nanolaminate heterostructure consists of alternating atomically thin layers of $Al_2O_3$ and Ir metal[49] and was prepared using atomic-layer deposition (see the Supplementary information for fabrication details). We showed that the presence of the nanolaminate increases the detected current in the probe compared to a bare dielectric medium, see the Supplementary information for this comparison. The coating itself limits the depth of the active volume that generates the ultrafast currents to its thickness, which was 227 nm. Next, the lateral size of the active volume is limited by the shape of the electrodes. The horizontal size (laser polarization direction) is bounded by the distance between the electrodes which was designed to be 570 nm. The vertical size (perpendicular to laser polarization) is defined by the width of the electrodes. To push this resolution to the maximum we opted for a triangular shape of the electrodes with a sharp vertex, so that the vertical dimension of the active volume is shorter than the horizontal one, see Fig. 1b. Overall, taking the upper limit for the vertical size same as the distance between electrodes: 570 nm, we end up with an active volume, i.e., volumetric resolution of $0.1 \mu m^3$.

### Acquisition method
The CEP probe was moved in steps of 3 or 5 μm along the optical axis and 250 nm in the polarization plane. As the CEP probe is being moved step-by-step in the beam being tested, we acquire the magnitude of the current $J_O$ at phase $\varphi_J$ as a function of spatial coordinates $x$, $y$, and $z$. The speed of the CEP scan was limited by the speed of the moving stage and the relationship between the bandwidth of the preamplifier and $f_{CEO}$. In this work, the preamplifier's (FEMTO® Messtechnik GmbH, DLPCA-200) bandwidth was 1 kHz, $f_{CEO}$ was 1 kHz and 30 ms settling time (time constant) of the lock-in amplifier, which could theoretically allow detection speeds up to 30 samples/s. In the measurements presented here, the limit was the moving stage as its motion speed was about 2 Hz. If the $f_{CEO}$ was higher and the motion speed would be improved one could achieve an effective real-time feedback loop.

### Laser beam focusing
To reach the required on-target intensity the beam was expanded by a reflective off-axis parabola telescope to 6 mm FWHM in the intensity and then focused with an off-axis parabola of 15 mm focal length. This gives a theoretical low limit for the waist of the focus of 0.75 μm (intensity FWHM). Due to the wide spectrum and aberrations introduced by the use of parabolic mirrors in the beamline, the achievable size of the focal spot in terms of waist $w_O$ was below 1.7 μm. The size was determined with a knife-edge method, see the Supplementary information.

### Theoretical range of scanning
The amount of maximum intensity available will determine the size of volume that can be scanned. However, an estimate can be done in terms of length normalized to beam parameters $w_O$ and $z_r$, being beam waist and Rayleigh length, respectively. In the best-case scenario, assuming conditions of our experiment and a Gaussian spatial profile

of the beam, we can expect CEP sensitivity of the probe up to radius $1.6w_O$ in the laser focal plane. Concerning the sensitivity range along the laser propagation axis, we can expect a theoretical threshold of the on-axis CEP signal up to $12z_r$.

### Definition of $g$, $\gamma$, and $C_r$
The analytical model used in this work uses parameters $g$ and $\gamma$ to describe the chromatic aberrations of the spectrally broad laser beam:

$$g = 1 + \frac{2w_0'}{w_0}\omega_0, \gamma = \frac{f'}{z_r}\omega_0 \qquad (1)$$

where $w_O$ is the beam waist, $f$ is the focal position, $z_r$ is the Rayleigh length, and the prime sign stands for a derivative with respect to frequency $\omega$ evaluated at the central frequency, which is in our case $\omega_O = 2.356$ rad/fs. In addition, the $\Delta CEP(x,y,z)$ is also a function of the spectral phase, i.e., in the first approximation chirp. Throughout this work we use a relative chirp defined as $C_r = 2\ln(2)C/\tau^2$, where $C$ is the chirp. For a pulse with a spectrum $E(\omega) = \sqrt{S(\omega)}\exp(-i\varphi(\omega))$, $C = 2\varphi''(\omega_0)$, i.e., the chirp is equivalent to the group velocity dispersion and can be estimated from the second derivative of spectral phase at the central frequency.

### Model for calculating CEP spatial maps
To model the $\Delta CEP(x,y,z)$ spatial map, we used an analytical formula from[45]:

$$\Delta CEP(r,z) = -\tan^{-1}\left(\frac{z}{z_r}\right) + \frac{1 - 2\frac{r^2}{w^2(z)}}{1 + \left(\frac{z}{z_r}\right)^2} \times \left[G\left(\frac{z}{z_r}\right) + \Gamma\left(\frac{z}{z_r}\right)^2\right]$$
$$+ \left[\Gamma + 2C_r\right]\frac{r^2}{w^2(z)} \qquad (2)$$

where

$$G = g - 2\gamma C_r, \Gamma = \gamma + 2gC_r \qquad (3)$$

with a radial coordinate $r^2 = x^2 + y^2$ and $z$ being the distance along the optical axis. We use the Eq. (2) in Figs. 3–5 as well as for selected cases of $C_r$ and $g$ in the Supplementary information. The model clearly shows how the parameters $g$, $\gamma$ or $C_r$ (representing the first approximation of the spectral characteristic of the laser beam) can modify the CEP evolution in space.

### Relation between pupil function and CEP
We showed that a shaping of the phase front of the pupil function with SLM can affect the CEP landscape. We would like to comment on the relationship between the pupil function and the CEP distributions around the laser focus. Although the SLM does not affect the CEP directly, it adds phase to the laser beam wavefronts which in turn is added to the focal phase distribution. Mathematically speaking, the distribution of the electric field in the focal plane is a Fresnel integral over the pupil function. Thus, shaping the pupil function has the potential to change the intensity and phase distribution about the focal plane. Clearly, the CEP phase is not just the phase of the wavefront, but it results from its relation to the pulse-front surfaces. Therefore, the complete relationship between the pupil function and the focal CEP phase requires demanding spatiotemporal modeling that is beyond the scope of this paper. Here, we decided to overcome this complexity and for the sake of proof-of-principle experiments, we do not aspire to prove the relationship between the pupil function and the focal CEP phase, but we intend to show that it is possible to realize a feedback loop where step-by-step optimization of a pupil function leads to a desired CEP phase distribution in the focus.

## Data availability

The data that support the findings of this study are available from the corresponding author upon request.

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

## Acknowledgements

We acknowledge support from a FET Open Grant of the EU (PetaCOM 829153) and from the National Research, Development and Innovation

Office of Hungary (Projects 137373, TKP-2021-NVA-04 and 2018-1.2.1-NKP-2018-00012). We thank Balázs Major for discussions. The development of Ir/Al$_2$O$_3$ heterostructures was supported by the German Research Foundation (DFG, Deutsche Forschungsgemeinschaft) Collaborative Research Center (CRC/SFB) 1375 "NOA—Nonlinear Optics down to Atomic scales"—project B3, number 398816777, and the Fraunhofer Society Attract Project (grant number 066- 601020), Fraunhofer IOF Center of Excellence in Photonics.

## Author contributions

V.H. and P.D. conceived and designed the experiments; V.H., B.F., and V.C. performed the experiments; V.H. and P.S. analyzed the data; Z.P., J.B., Z.W., P.P., and A.S. contributed with materials/analysis tools; V.H. and P.D. wrote the paper.

## Funding

## Competing interests

The authors declare no competing interests.
