## [Peer review file · Nature Communications]

REVIEWER COMMENTS

Reviewer #1 (Remarks to the Author):

The paper by Hanus et al presents two achievements: (i) the measurement of the CEP with a probe that works in ambient air and provides sub-micron spatial resolution. (ii) the manipulation of the (generalized) Gouy phase such that a relatively flat Gouy phase is achieved in the focus. Two respective approaches are described.

In my opinion, these results are worth publication in Nature Communication in principle although I have a number of questions and remarks. Some of these are minor, others are not.

1 l. 23: To my knowledge, the papers that established frequency comb spectroscopy and CE phase stabilization (and the respective Nobel prize ...) are Reichert et al., Opt. Comm 172, 59 (1999) and Jones et al., Science 288, 635 (2000)

2 l. 32: I would assume that phase-averaging is equally significant for HHG in gases. Why the restriction to nano-structured solids?

3 l. 47: What is meant by „single-beam probe“. Just a probe with high spatial resolution?

4 l. 54: „Tagging“ means that the CEP is measured for each and every pulse and the corresponding events of the (phase-dependent) experiment under consideration are „tagged“ with the CEP corresponding to the event. Sorting the events according to the corresponding CEPs then results in the phase-dependence. This is not what is done in [6]. In fact, [6] also does not use the word tagging.

5 l. 60: The authors should not claim „volumes of constant (frozen) CEP“. At best, there are volumes of approx. constant CEP. In fact, the variation of the Gouy phase in the focal volume (and thus the variation of the CEP) is a fundamental, i.e. an unavoidable consequence of the wave-nature of light. Accordingly, I would assume, there are quite tight limitations for manipulating the phase distribution in the focus, in particular under the constraint that the phase should be flat where the intensity is highest in the focus. In my opinion, these limitations should be discussed. After all, it is quite relevant to know how close the present results are to what is theoretically possible.

6 l. 75ff: The authors should spent one sentence (or two) on the differences of the present CEP probe to the one they presented in Optica in 2021.

7 l. 90: 1.3×10^{13} W/cm² ? Aren't there issues with optical damage?

8 l. 121: What is meant by point approximations?

9 l. 128ff: These two sentences are particularly awkward. (BTW: Quite a few others, too.)

10 l. 142: The text speaks of a change of Cr by 1.6. In Fig. 3, there are only data for Cr=0 and Cr=0.8. Maybe, this has to do with using two different definitions of Cr, namely $2 \cdot \log_2 C/\tau^2$ and $4 \cdot \log_2 C/\tau^2$?

11 Different measures are used as figures-of-merit in „freezing“ the CEP: When chirp is used, an area of approx. constant phase is used (l. 150), but for the SLM the volume of approx. constant phase (l. 211).

12 l. 180: I don't understand what is meant by „cup-shaped“. I can't recognize something reminiscent of a cup in Fig. 2

13 l. 195: Another particularly awkward sentence. I guess that the (magnitude of the) slope of CEP(z) is meant.

14 Fig. 5a: I would have expected that the laser beam has a phase distribution of cylindrical symmetry for $s=\infty$.

15 l. 202: The text says that the phase is not flat along the horizontal axis. But it is also not flat in vertical direction. According to Fig 5a, the largest slope occurs on the diagonal.

16 Fig. 5c: I understand what s means. However, I didn't find in the text by how much the phase differs in the V-shaped area and the rest of the beam.

17 Finally a question on the significance of the entire part on manipulating the (generalized) Gouy phase for experiments using amplified pulses: For pulses from an oscillator, the g-factor is rather small and the phase evolution along the optical axis even flatter than in the monochromatic case. This is different for amplified pulses, at least if they are compressed to few-cycle duration by the hollow-

core fiber method. The question arises whether the methods presented in the paper can be applied to these important situations and what one could expect, see also 5.

Reviewer #2 (Remarks to the Author):

In their manuscript, Hanus et al. provide a microscopic study of the spatial distribution of the carrier-envelope phase (CEP) of a few-cycle laser pulse within the confocal region. They achieve this measurement with a nanodevice based on two Au electrical contacts with a gap of order 500 nm at the surface of a nanolaminate of Al₂O₃ and Ir grown by ALD. By scanning the device through the confocal region, three-dimensional maps of the relative CEP are obtained based on the phase of the photoinduced current. The authors also use a spatial light modulator in the ultrashort pulse train to manipulate the CEP distribution, showing the potential to also optimize it for special tasks in related experiments. This experiment reports on interesting phenomena related both to very general principles of optics as well as specific tasks related to subcycle manipulation of matter. Therefore, it should be interesting to read for both general and specialized scientific readerships. Therefore, the content of this study is eligible for publication in Nature Communications.

When reading through the manuscript, though, the (admittedly somewhat specialized) referee noticed some inconsistencies derived e.g. from special terminology the authors used to setup the paper. In the following, the tendentially irritating aspects are summarized. The authors are motivated to work on these items in a revised version in order to make their study more directly accessible and precise:

(a) In line 55 of the main manuscript, the authors state that "exploit light-field-driven CEP-sensitive ultrafast currents in a solid medium". No further specification of this medium is found in the main text and figure captions. In Fig. 1, only "Al₂O₃:Ir" is indicated within the green frame which would mean "aluminum oxide doped with iridium". One has to read all the way through to the Methods section to learn that we are dealing here with a nanolaminate of alumina and iridium grown by atomic-layer deposition (ALD). Still, the curious reader is wondering what the physics is behind the photoinduced current? Information on this question is found only in the Supplementary where the authors state "We surmise that these advantageous properties of this material stem either from its large nonlinear susceptibility $\chi^{(3)}$ [1] or from a conduction band that is not empty." It would be great for the reader to have some of this information already within the main part of the paper.

(b) Already in the abstract (line 14) and at several points in the later manuscript (line 97), the authors state that they demonstrate a "carrier-envelope phase (CEP) scanning probe" or equivalent. Initially, the referee was wondering how this is achieved because e.g. the metallic component in their medium or plasmonic resonances of the contacts might induce some a-priori unknown shifts between the free-space CEP of the pulse train at a specific spot and the phase of the detected photocurrent. See, e.g. Nature Phys. 16, 341 (2020) for an investigation of such effects. Of course, this discrepancy is resolved later on in the manuscript (e.g. around line 84) where it becomes clear that the device only measures the relative CEP offset instead of an absolute CEP value. Nevertheless, the authors should do their best to avoid such ambiguities throughout the paper.

(c) In the same sense, also the expression "CEP freezing" (already in the abstract line 17 and similarly at later paragraphs, e.g. lines 60, 133...) contributes to this potential misunderstanding: First, it suggests some absolute measurement of CEP which the authors do not accomplish. But on top of that, the term "frozen" suggests some stationary character which is also not present here due to the finite carrier-envelope offset frequency f_{CEO} of the pulse train studied. Instead, what is meant seems to be a CEP which is constant over a certain region of space and the authors are motivated to make sure this aspect is understood correctly as early as possible in the paper.

(d) This referee suggests to replace the adjective "on-air" with e.g. "ambient-air" throughout the manuscript in order to sustain maximum scientific precision.

(e) One of the fascinating aspects of the present work is this direct insight into the phase evolution of a focused pulse in space. For example, around line 119 the authors state "This contrasts to a purely Gouy phase shift, which shows a symmetric and larger drop of CEP value." These aspects are discussed in detail in the rest of the paper. What seems to be somewhat omitted, though, is the

aspect that deviations from a theoretical model picture could also result from non-ideal properties of the probe itself. E.g. what about the influence of a non-ideal CEP response of the device: there is definitely a finite spatial resolution which is in the same scale as the spatial patterns discussed here. Is the photocurrent really perfectly symmetric for CEPs differing by π , i.e. how perfect is the spatial symmetry of device? Unfortunately, the SEM micrograph in Fig. 1(b) is obscured by (presumably artistic) red dot in decisive region of the device. Also, the highly nonlinear amplitude response of the device should distort the comparison to the theoretical models where this aspect is not taken into account. It would be great if the authors could include these items more clearly into their paper in order to increase its quantitative character.

(f) In lines 142/143 the authors state "The experiment is limited in the range of glass that can be added as the laser pulse gets longer and consequently signal from the probe weaker.". Note that this detrimental effect may be avoided by changing the CEP in a region where the spectral bandwidth of the pulse is still limited, as possible e.g. in ultrabroadband fiber laser technology (compare Ref. 14).

(g) In lines 228/229, the authors state "We anticipate that our technique is compatible with a wide range of laser systems as long as they are phase-locked to a nonzero f_{CEO} ...". Is it possible with this device to obtain and measure also a DC current when $f_{\text{CEO}} = 0$? There could be amplitude modulation with e.g. a chopper wheel which also enables lock-in detection. Remember that f_{CEO} is only 1 kHz in this study (Methods section!) which is easily compatible with mechanical options of modulation.

The somewhat lengthy items above are meant only as a support for the authors to improve their manuscript by rendering it as scientifically sound as possible, also in terms of terminology. Clearly, the great results and sub-wavelength insights into few-cycle electrodynamics merit publication in Nature Communications as soon as these issues are taken care of.

Reviewer #3 (Remarks to the Author):

The authors present a device to scan the CEP of a laser focus in three dimensions under ambient conditions. First results including some degree of control of the CEP in a laser focus are demonstrated. The work described in the manuscript is highly relevant to current research because the field distribution in a laser focus plays the(!) crucial role in the interaction of short laser pulses with nanostructures. The manuscript relates the work well to other work that has achieved similar results. The recently published paper Bionta, M. R. et al. On-chip sampling of optical fields with attosecond resolution. Nat. Photonics 15, 456-460 (2021) would have been additionally appropriate to reference. Unfortunately, the extensive discussion of the CEP in a laser focus and attempts to control it is at the expense of describing and characterizing the measurement method itself: It remains unclear how a macroscopic current is obtained microscopically from the interaction of the laser field with the nanostructure and how the CEP change is extracted from it. It is mentioned that the local intensity has an influence on the measurement, but this is not verified in the manuscript itself. Also, the influence of polarization changes in the focus on the measurement is not examined. The spatial resolution is given in terms of a volume, lateral values and their determination would be more helpful here. Accordingly, the second part of the paper discusses measurements where the reliability of the data is unclear to the reader. I would highly recommend to focus more on a thorough characterization of the measurement method before investigating a complex laser focus and drawing conclusions from it.

Additional questions and comments that arose while reading the paper:

Line 95: What is the value of w_0 and what is the resulting number of usable pixels/voxels?

How are Gouy phase shifts and CEP changes separated?

Line 130: Meaning of C_r unclear to the reader. How do the found negative chirp and a separate d-scan measurement fit? Please state the numbers.

Line 142: C_r should change by 1.6, line 147 only states a difference of 0.8. Why?

Line 149: Does this include Gouy phase changes?

Line 172: What do g and γ mean, why are they affected?

Reply to Reviewer 1

We thank Reviewer 1 for the time and effort spent with the review and for finding our work suitable for publication in Nature Communications journal. Furthermore, we are grateful for the detailed comments and suggestions which we address in detail in the following. Changes in the manuscript are then highlighted in light blue and listed at the end of the letter.

Comment 1:

I. 23: To my knowledge, the papers that established frequency comb spectroscopy and CE phase stabilization (and the respective Nobel prize ...) are Reichert et al., Opt. Comm 172, 59 (1999) and Jones et al., Science 288, 635 (2000)

Reply:

We thank for the notice and suggestion. We updated the references.

Comment 2:

I. 32: I would assume that phase-averaging is equally significant for HHG in gases. Why the restriction to nano-structured solids?

Reply:

Indeed, the phase is important in gases as well, However, here we wanted to list categories of experiments that have not demonstrated CEP sensitivity so far. To our best knowledge, the CEP sensitivity in case of gases was already demonstrated, while in solids not. However, there is a potential that CEP distribution scanning and control could also impact HHG in gases.

Comment 3:

I. 47: What is meant by „single-beam probe“. Just a probe with high spatial resolution?

Reply:

We referred to a type of CEP distribution measurement that does not need an interferometric setup. Such an interferometric setup was demonstrated for example B. Major et al. Appl. Opt. 54, 10717 (2015). We changed the wording to “non-interferometric” for better clarity.

Comment 4:

I. 54: „Tagging“ means that the CEP is measured for each and every pulse and the corresponding events of the (phase-dependent) experiment under consideration are „tagged“ with the CEP corresponding to the event. Sorting the events according to the corresponding CEPs then results in the phase-dependence. This is not what is done in [6]. In fact, [6] also does not use the word tagging.

Reply:

Indeed, the use of tagging is not appropriate as the method in [6] does not allow measurement of CEP of single pulses and the pulses in the train needs to vary CEP by π from one to another. We changed “CEP tagging device” for “CEP measurement device” to avoid misunderstanding.

Comment 5:

I. 60: The authors should not claim „volumes of constant (frozen) CEP“. At best, there are volumes of approx. constant CEP. In fact, the variation of the Gouy phase in the focal volume (and thus the variation of the CEP) is a fundamental, i.e. an unavoidable consequence of the wave-nature of light. Accordingly, I would assume, there are quite tight limitations for manipulating the phase distribution in the focus, in particular under the constraint that the phase should be flat where the intensity is highest in the focus. In my opinion, these limitations should be discussed. After all, it is quite relevant to know how close the present results are to what is theoretically possible.

Reply:

The Reviewer is right that the CEP values cannot be strictly constant. For measured values one needs to set an interval where a potentially varying value is considered to be constant. Later in the manuscript, we explicitly define an interval of 10° width: (-5°, +5°).

As the reviewer points out correctly, the variation of the CEP is linked to the wave-nature of the light and in the case of single-wavelength CW lasers that are perfect Gaussian beams, it is the Gouy phase that governs the phase evolution. However, the CEP is a quantity of pulsed light, which has a wide spectrum by definition. The spectrum can have very complex spatial and temporal characteristics that have an impact on the CEP spatial distributions. For example in [1], the influence of first derivative of waist (parameter g) and beam size (parameter γ) as a function of the wavelength was theoretically studied. Also the spectral chirp, C_r , was found to cause significant deviations from the Gouy phase along the optical axis and, interestingly, it also causes a curvature in the radial direction in the region of the focus, where the intensity is the highest, see Fig.4 (a3,b3) of revised manuscript. Yet a comprehensive framework for broadband laser pulses that would analyze the octave spanning spectrum (as in the case of few-cycle lasers) was not yet, to the best of our knowledge, developed nor published.

To make the influence of the parameters g , γ and C_r clearer, we added the formula from [1] used for calculating the 3D CEP maps to the *Methods* section. We would like to refer the Reviewer also to the Supplementary Information, FigSI. 6, where we evaluate the eq. (2) for multiple parameters.

The limitations of the CEP sculpting are set by the possibilities of modulating the spectral phases in space and time. The modulation with SLM device used in this manuscript relies on the mismatch between the phase velocities of the spectral components, imposing a range of wavefront curvatures as a function of wavelength – a chromatic aberration. This way g and γ are influenced and consequently CEP distribution. Such a chromatic aberration effect on CEP achieved with a lens was theoretically analyzed also in [1].

Please see also the answer to comment 17 for a discussion of application limitations to the post-compressed pulses. We have added an analysis of the SLM in introducing the chromatic aberrations in the Supplementary information.

Comment 6:

I. 75ff: The authors should spent one sentence (or two) on the differences of the present CEP probe to the one they presented in Optica in 2021.

Reply:

We added a sentence in the Introduction section to highlight the differences between the samples/probe

Comment 7:

I. 90: 1.3×10^{13} W/cm² ? Aren't there issues with optical damage?

Reply:

During the target evolution we did encounter some issues with optical damage, but our last iterations of the substrate and electrodes were resistant to damage up to this intensity value. We would like to refer the Reviewer to for example A. Korobenko et al., Nat. Commun. 12, 4981 (2021) [2], that illuminates solid targets at similar conditions. The damage starts in their case slightly higher at 1.5×10^{13} W/cm². We added a sentence mentioning that no damage happens to the probe at 1.3×10^{13} W/cm².

Comment 8:

I. 121: What is meant by point approximations?

Reply:

We apologize for this vague formulation. We improved the sentence being more specific and mentioned right away that g , γ , and C_r (chirp), are the first derivatives of beam waist, focal position, and spectral phase, respectively, with respect to frequency evaluated at the central frequency.

Comment 9:

I. 128ff: These two sentences are particularly awkward. (BTW: Quite a few others, too.)

Reply:

We thank for the notice. We have reformulated the paragraph.

Comment 10:

I. 142: The text speaks of a change of C_r by 1.6. In Fig. 3, there are only data for $C_r=0$ and $C_r=0.8$. Maybe, this has to do with using two different definitions of C_r , namely $2 \cdot \log^2 C/\tau^2$ and $4 \cdot \log^2 C/\tau^2$?

Reply:

Indeed, we were inconsistent in using the C_r definitions. We have updated the manuscript to use definition $C_r = 2 \cdot \log^2 C/\tau^2$ everywhere. As a consequence the model evaluation had to be adjusted and some values of g and γ were updated as well as corresponding plots in Figures 3-5 (revised version numbering).

Comment 11:

Different measures are used as figures-of-merit in „freezing“ the CEP: When chirp is used, an area of approx. constant phase is used (I. 150), but for the SLM the volume of approx. constant phase (I. 211).

Reply:

In coherence with other reviewer's remark we stopped using „freezing“ in the paper.

The apparent inconsistency of introducing two figures of merit was to pinpoint different effects, which in the case of chirping was the flattening of the phase in a plane, while the SLM approach gave rather a volume flattening. Both of which can be useful in different situations.

We removed the quantitative evaluation of the area and we stick to just a qualitative formulation that the area of flat CEP in the vicinity of the intensity maximum increased. This way we will not confuse the reader with using two types of figure of merit.

Comment 12:

I. 180: I don't understand what is meant by „cup-shaped“. I can't recognize something reminiscent of a cup in Fig. 2

Reply:

We have removed this unclear expression.

Comment 13:

I. 195: Another particularly awkward sentence. I guess that the (magnitude of the) slope of CEP(z) is meant.

Reply:

We reformulated the paragraph hoping for better clarity for readers.

Comment 14:

Fig. 5a: I would have expected that the laser beam has a phase distribution of cylindrical symmetry for $s=\infty$.

Reply:

That would be indeed true for a perfect TEM₀₀ Gaussian beam. Concerning our study, we present measurements on real beams that always contain some aberrations as for example astigmatism. That is one of the reasons why our beam is elliptical in the focus. The amount of aberrations can change on daily basis as the alignment of the focusing parabola needs to be done with sub-mrad precision. The example of characterization of the focus is displayed in FigSI. 5 in the Supplementary Information.

Comment 15:

I. 202: The text says that the phase is not flat along the horizontal axis. But it is also not flat in vertical direction. According to Fig 5a, the largest slope occurs on the diagonal.

Reply:

We removed the statement about the horizontal axis and reduced the description on “not a flat CEP response”.

Comment 16:

Fig. 5c: I understand what s means. However, I didn't find in the text by how much the phase differs in the V-shaped area and the rest of the beam.

Reply:

This important value we unintentionally omitted. The phase jump on the correction pattern of SLM is set to π . We inserted this value to the text, when presenting the V-shape pattern.

Comment 17:

Finally a question on the significance of the entire part on manipulating the (generalized) Gouy phase for experiments using amplified pulses: For pulses from an oscillator, the g -factor is rather small and the phase evolution along the optical axis even flatter than in the monochromatic case. This is different for amplified pulses, at least if they are compressed to few-cycle duration by the hollow-core fiber method. The question arises whether the methods presented in the paper can be applied to these important situations and what one could expect, see also 5.

Reply:

To the best of our knowledge the question of how g differs for different types of lasers was not yet sufficiently explored. More detailed studies with CEP characterization were published for an amplified beam with hollow-core fiber post-compression in papers by Hoff et al. [3,4], where g was determined to be around -2. In our oscillator case we arrived to values +0.2 to +0.4, so, indeed, the absolute value of g for oscillators seems to be lower than that from the hollow-core fiber compressor. However, the phase evolution along the optical axis is not governed by g only, but by γ and chirp as well. As g is considered to be difficult to change and chirping can be limited by the desired pulse duration, it is only γ that can be effectively used to tune the CEP profile. It was proposed by Porras [1] that γ is changed with lenses. This is what motivated us to use SLM as it partially mimics a use of a lens as it introduces different phase shifts for different wavelengths. We show in the article (comparing Fig. 3 and Fig. 5) that the application of the lensing pattern did decrease γ from 0.3 to -0.4 as obtained from the matching model situations.

Let us discuss the possibilities to apply the SLM method for few-cycle lasers based on hollow-core fiber postcompression. We think that the key is to address the chromatic aberrations presented in the fiber-output beam as they also influence parameters g and γ . Therefore, SLM should be able to manipulate certain range of color-dependent phase shifts across the size of the beam. As a representative example, we can consider laser beam characteristics presented by Alonso et al. [5]. We extracted Fig. 5(b) of that article in the figure below, panel (b). The figure shows the wavefront across the beam radius for two selected colors: 600 and 900 nm. One can see that the blue components are more diverging and the range of the wavefront curvature reaches about 20 radians with a difference between the color components amounting to 7.5 radians. Applying a curved pattern to the SLM would cause a retardation while in the center the blue components would be more retarded than the red ones, see panel (a) (note that in this example, the pattern is an inverted version of the one used in Fig. 5 of our main manuscript). As a consequence, the chromatic aberration can be reduced by the application of a curved pattern on the SLM as the SLM induces enough phase shift to invert the difference between the color components, see panel (c), where the phase difference (black line) between color components is flatter than in (b). At the same time SLM pattern can be sculpted further in order to achieve custom CEP spatial distribution in a similar manner which is presented in Fig. 6 of the main manuscript. In case some system possesses higher aberrations than considered in this example, one would need to find ways to precompensate them with other means, e.g. with lenses.

We added a few sentences discussing the applicability to hollow-core fiber compressor outputs in the manuscript. We added a discussion about the SLM capabilities to the Supplementary information.

Fig. 1 Proposition for compensating the chromatic aberration of typical output of a hollow-core fiber compressor laser system. (a) Lensing pattern on SLM with high pixel value in the middle retards the 600-nm component (blue line) more than the 900-nm one (red line), while at the outer part of the beam the retardation are equal. (black line) shows the difference between these two components. (b) A representative wavefront of a hollow-core fiber compressor output. (c) Wavefront resulting from the addition of the SLM-induced retardation and the fiber output. The phase difference at the outer parts of the beam were decreased by the action of the SLM.

List of bigger changes to the manuscript (highlighted with blue font color in the manuscript text)

- Introduction, last paragraph: Reformulated the relation to the previously published paper in Optica in 2021; Comment that no vacuum tunneling is present.
- Paragraph 2.2: notion of no damage to the sample.
- Paragraph 3.2: g and gamma description added.
- Paragraph 3.2: Awkward sentences reformulated.
- Paragraph 4.2: Applicability to hollow-core fiber systems
- Paragraph 4.4: Awkward sentences reformulated.
- “on-air” changed to “ambient-air”

Reply to Reviewer 2

We thank Reviewer 2 for the time and effort spent with our manuscript and for finding that our results are worth publishing in Nature Communication. Furthermore, we are grateful for the comments which we reply to in detail below. Changes in the manuscript are then highlighted in orange and listed at the end of the answers.

Comment (a):

(a) In line 55 of the main manuscript, the authors state that "exploit light-field-driven CEP-sensitive ultrafast currents in a solid medium". No further specification of this medium is found in the main text and figure captions. In Fig. 1, only "Al₂O₃:Ir" is indicated within the green frame which would mean "aluminum oxide doped with iridium". One has to read all the way through to the Methods section to learn that we are dealing here with a nanolaminate of alumina and iridium grown by atomic-layer deposition (ALD). Still, the curious reader is wondering what the physics is behind the photoinduced current? Information on this question is found only in the Supplementary where the authors state "We surmise that these advantageous properties of this material stem either from its large nonlinear susceptibility $\chi^{(3)}$ [1] or from a conduction band that is not empty." It would be great for the reader to have some of this information already within the main part of the paper.

Reply:

We thank the referee for the notice. We added more description after the cited sentence about the active material and its relation to $\chi^{(3)}$.

Comment (b):

Already in the abstract (line 14) and at several points in the later manuscript (line 97), the authors state that they demonstrate a "carrier-envelope phase (CEP) scanning probe" or equivalent. Initially, the referee was wondering how this is achieved because e.g. the metallic component in their medium or plasmonic resonances of the contacts might induce some a-priori unknown shifts between the free-space CEP of the pulse train at a specific spot and the phase of the detected photocurrent. See, e.g. Nature Phys. 16, 341 (2020) for an investigation of such effects. Of course, this discrepancy is resolved later on in the manuscript (e.g. around line 84) where it becomes clear that the device only measures the relative CEP offset instead of an absolute CEP value. Nevertheless, the authors should do their best to avoid such ambiguities throughout the paper.

Reply:

We are sorry that we could not convey the information more clearly. Indeed, the probe allows to measure the CEP only in a relative manner. We have updated the figure caption to highlight that we measure "spatial change" of CEP and we introduced Δ CEP in figures and text. Concerning Ludwig's paper about tunneling in the nanoscale gap, we included a note highlighting that electrodes are far enough that the tunneling cannot happen. The metallic component in the active material forms rather continuous layers instead of nanoparticles and the linear optical properties do not show enhancement in absorption, therefore, plasmonic resonances are not expected in the VIS-IR spectral range.

Comment (c):

In the same sense, also the expression "CEP freezing" (already in the abstract line 17 and similarly at later paragraphs, e.g. lines 60, 133...) contributes to this potential misunderstanding: First, it suggests some absolute measurement of CEP which the authors do not accomplish. But on top of that, the term "frozen" suggests some stationary character which is also not present here due to the finite carrier-envelope offset frequency f_{CEO} of the pulse train studied. Instead, what is meant seems to be a CEP which is constant over a certain region of space and the authors are motivated to make sure this aspect is understood correctly as early as possible in the paper.

Reply:

As the Reviewer suggests, we removed the term "freezing" and replaced it with flattening at the corresponding places in the article. We now formulate that we measure CEP changes and introduce Δ CEP notation.

Comment (d):

This referee suggests to replace the adjective "on-air" with e.g. "ambient-air" throughout the manuscript in order to sustain maximum scientific precision.

Reply:

We adopted this suggestion all through the article. Thank you.

Comment (e):

One of the fascinating aspects of the present work is this direct insight into the phase evolution of a focused pulse in space. For example, around line 119 the authors state "This contrasts to a purely Gouy phase shift, which shows a symmetric and larger drop of CEP value." These aspects are discussed in detail in the rest of the paper. What seems to be somewhat omitted, though, is the aspect that deviations from a theoretical model picture could also result from non-ideal properties of the probe itself. E.g. what about the influence of a non-ideal CEP response of the device: there is definitely a finite spatial resolution which is in the same scale as the spatial patterns discussed here. Is the photocurrent really perfectly symmetric for CEPs differing by π , i.e. how perfect is the spatial symmetry of device? Unfortunately, the SEM micrograph in Fig. 1(b) is obscured by (presumably artistic) red dot in decisive region of the device. Also, the highly nonlinear amplitude response of the device should distort the comparison to the theoretical models where this aspect is not taken into account. It would be great if the authors could include these items more clearly into their paper in order to increase its quantitative character.

Reply:

Indeed, there could be some deviations that result from non-ideal properties of the probe. We are fully aware of this possibility and that is why we have performed the study of the phase change dependence on the laser intensity, see FigSI. 4(a), that showed very little dependence. We also corrected for this effect in our measurements, as described in the Supplementary information. Thus, we believe that the nonlinear response in the current amplitude was compensated by the correction obtained from the electric field scans, visualized in FigSI. 4(b). Moreover, it can be shown that the sensitivity to CEP change is high and the current changes its direction together with the CEP. To make this clear, we have reformulated the manuscript and expanded a section in the Supplementary Information to show how the measured current oscillates with the change of CEP, see the updated FigSI. 2(b,c). We have also added a new figure (Fig.2) to the main manuscript which was previously part of the Supplementary Information.

Concerning the finite spatial resolution, we believe that it is determined by the size of the electrodes as we were able to sample the beam focal profile with a sufficient resolution, see new Fig. 2(a). The spatial profile of the current was smaller ($w_0 = 1.1 \mu\text{m}$) than the size of the beam measured with a knife edge method ($w_0 = 1.7 \mu\text{m}$) as expected assuming a nonlinear response to the intensity. This makes us assume that the spatial resolution is determined only by the distance between the electrodes – in this case: $0.57 \mu\text{m}$.

We modified the SEM micrograph in Fig. 1(b) so that the electrodes are better visible.

Comment (f):

In lines 142/143 the authors state "The experiment is limited in the range of glass that can be added as the laser pulse gets longer and consequently signal from the probe weaker.". Note that this detrimental effect may be avoided by changing the CEP in a region where the spectral bandwidth of the pulse is still limited, as possible e.g. in ultrabroadband fiber laser technology (compare Ref. 14).

Reply:

We thank referee for this insightful remark. Although the possibility to modulate CEP with glass insertion without the detrimental effect of chirping is intriguing, here, the chirp introduction was intentional as it is the chirp that changes the CEP distribution. We wanted to highlight that CEP effects are weaker as the pulse gets longer, which is a general property of CEP dependent phenomena. We reformulated the sentence to better communicate our message.

Comment (g):

In lines 228/229, the authors state "We anticipate that our technique is compatible with a wide range of laser systems as long as they are phase-locked to a nonzero f_{CEO} ...". Is it possible with this device to obtain and measure also a DC current when $f_{\text{CEO}} = 0$? There could be amplitude modulation with e.g. a chopper wheel which also enables lock-in detection. Remember that f_{CEO} is only 1 kHz in this study (Methods section!) which is easily compatible with mechanical options of modulation.

Reply:

Unfortunately, we are not aware of a possibility of DC current measurement at $f_{\text{CEO}} = 0$. We performed some experiments to do so and we found a limitation in terms of the very high background current (about 4-5 orders of magnitude higher) that is CEP-independent and is triggered at the laser repetition rate. For that reason alternative ways of detection as e.g. an amplitude modulation with chopper does not work without further improvements to increase signal-to-noise ratio. However, one could think about a mechanical CEP modulation for systems that do not have the capability to phase lock at $f_{\text{CEO}} \neq 0$. That can be easily achieved with a piezo dithering of a glass wedge at some reference frequency.

We thank the referee for this very good suggestion and we added a notion in the summary that mechanical modulations are the option for systems that lock to $f_{\text{CEO}} = 0$. We also added the value of 1 kHz of lockin detection to the main text.

List of bigger changes to the manuscript (highlighted with orange font color in the manuscript text)

- Introduction, last paragraph: Reformulated the relation to $\chi^{(3)}$.
 - Paragraph 4.1: Chirping range comment.
 - Paragraph 5.1: application to lasers that phase lock at $f_{\text{CEO}} \neq 0$.
 - New Figure 2: Part (c) of Fig.1 was moved in Fig. 2. Added panels describing the principle and robustness of the probe.
 - Notations changed in text and figures: CEP -> ΔCEP
 - on-air -> ambient-air
-

Reply to Reviewer 3

We thank Reviewer 3 for the time and effort spent with the review. Furthermore, we are grateful for the comments which we reply to in detail in the following. Changes in the manuscript are then highlighted in green and listed at the end of the letter.

Comment 1:

The recently published paper Bionta, M. R. et al. On-chip sampling of optical fields with attosecond resolution. Nat. Photonics 15, 456-460 (2021) would have been additionally appropriate to reference.

Reply:

We added this recommended reference to an appropriate place in the manuscript, where we discuss the CEP detection using the nanotunneling.

Comment 2:

... the extensive discussion of the CEP in a laser focus and attempts to control it is at the expense of describing and characterizing the measurement method itself: It remains unclear how a macroscopic current is obtained microscopically from the interaction of the laser field with the nanostructure and how the CEP change is extracted from it.

Reply:

Indeed, we have omitted some details about the CEP-dependent current detection technique. This was done intentionally as we wanted to focus the manuscript on the capabilities of the 3D CEP scanning technique, and the measurement method was already described in detail in our previous publication: V. Hanus et al., Optica 8, 570 (2021) [6]. That said, we agree with reviewer's concern that this can render the method description incomplete. In the main manuscript we have reformulated the introduction of the CEP probe design in order to highlight that the difference from the method in V. Hanus, et al., Optica 8, 570 (2021) lies only in the shape of electrodes and the used active medium. We also added a new Figure 2 with details about the CEP change measurement. As the Nature Comm. paper length is limited we kept some details of the method in the Supplementary Information, where we show how the lock-in current changes its sign as a function of CEP change, as well as the dependence of the current magnitude and phase on the pulse chirping, see FigSI. 2(b,c).

Comment 3:

It is mentioned that the local intensity has an influence on the measurement, but this is not verified in the manuscript itself.

Reply:

It is true that as the local intensity changes, the measured phase also changes slightly. This dependence was measured and quantized, see a new Fig. 2(c,e). We found out that the dependence is small (7.5°nm/V) and we compensated it for as described in the Supplementary Information. We have updated the text in the main manuscript to make the link to the Supplementary Information clearer.

Comment 4:

Also, the influence of polarization changes in the focus on the measurement is not examined.

Reply:

We thank the reviewer for this notice. The polarization influence can be analyzed from two aspects: The respective orientation of the probe and laser beam polarization, and the ellipticity state. Indeed, the initial ellipticity of the beam can cause various spatio-temporal effects around the focus (as analyzed theoretically in larger detail by Major et al. [7]). In this work, we do not expect anything like this, as our

laser beam polarization is linear with a high contrast (measured extinction with a polarizer was 0.2 %) and as we measure a free space propagation, we do not expect any mixing of perpendicular electric field components that would cause an elliptic polarization.

The focus of this paper are the fundamental effects observable with linear polarization. From this point of view, the respective orientation of the probe and the polarization vector E matters. The orientation of the probe is determined by a vector p , which is between the apices of the electrodes over the gap, see Fig. 1(b) for the geometry. We have examined the dependence of the CEP-dependent current on the angle θ between p and E , and we have noticed a decrease of the current with $\cos(\theta)$. This dependence can be useful for further theoretical investigation of the origin of ultrafast currents, but the detailed analysis is out of the scope of this paper since here we present our CEP scanner optical chip device and focal volume CEP characterization. However, we will perform systematic measurements on polarization effects and publish our findings in a separate paper.

We highlighted in the main manuscript the orientation of the polarization with respect to the probe. We added a characterization of the polarization state and the current-orientation dependence in the Supplementary Information, see FigSI. 10.

Comment 5:

The spatial resolution is given in terms of a volume, lateral values and their determination would be more helpful here.

Reply:

We have added a value for the lateral resolution. To keep the manuscript concise, we choose to keep the determination of the volume resolution in Methods section as it was.

Comment 6:

Accordingly, the second part of the paper discusses measurements where the reliability of the data is unclear to the reader. I would highly recommend to focus more on a thorough characterization of the measurement method before investigating a complex laser focus and drawing conclusions from it.

Reply:

We thank the Reviewer for this suggestion. We have created a new figure (No. 2) in the main manuscript that more clearly underpins the validity of our approach of measuring CEP spatial distributions. Now, Fig. 2 shows: The spatial response of the probe (panel (a)); correspondence of the CEP pulse train offset to the measured phase; discusses the dependence/independence of the phase and the current magnitude (c,e), and finally, it shows also the temporal stability. In the main manuscript we have highlighted these features and a more detailed analysis is now provided in the Supplementary material, where one can find additionally a change of the in-phase lock-in component J_{\cos} as a function of change of CEP implemented by a glass insertion, FigSI. 2. We hope that these prove the validity of our method and CEP scans.

Comment to Line 95:

What is the value of w_0 and what is the resulting number of usable pixels/voxels?

Reply:

We added the value w_0 . We thank the Reviewer for the suggestion about pixels/voxels and understand the need of presenting the useful range of the scans. We think that using the voxel number would be less informative than using the value of the volume or area. The number of voxels would need to be complemented with the 3D size of the voxel and the number of pixels. Therefore, we choose to present the useful area or volume of the scan in μm^2 or μm^3 . We now present in the main manuscript a value of $6.9 \mu\text{m}^2$ being a useful scan area when discussing Fig. 2 that presents capabilities of the CEP scanning probe and a useful scanned volume of $84 \mu\text{m}^3$ when discussing Fig. 3.

Comment to Line 130:

What is the value of w_0 and what is the resulting number of usable pixels/voxels? Please state the numbers.

Reply:

We added the value of w_0 (previously it was only in the caption of the figure) and the measured volume. The value corresponding to the situation in Fig. 3 is $w_0 = 1.7 \mu\text{m}$.

Comment to Line 142:

C_r should change by 1.6, line 147 only states a difference of 0.8. Why?

Reply:

We found out that we were inconsistent in using two C_r definitions differing by a factor 2. We removed the inconsistency and updated the manuscript to use definition $C_r = 2 \cdot \log_2 C / \tau^2$ everywhere. As a consequence the model evaluation had to be adjusted and some values of g and γ were updated as well as corresponding plots in Figures 3-5.

Comment to Line 149:

Does this include Gouy phase changes?

Reply:

Here, we would like to refer the Reviewer to equation (2) that we included in the Methods section of the revised manuscript. Equation (2) presents a model developed in [1] and illustrates that the manipulation of parameters as g , γ and chirp (C_r) results in an additive term to the Gouy phase term. Thus, from a certain point of view the Gouy phase does not change and the chirp only adds additional phase that has a spatial dependence.

Comment to Line 172:

What do g and γ mean, why are they affected?

Reply:

As an answer to this, when introducing g and γ in section 3, we added an explicit statement that these chromatic aberrations are first derivatives of the beam waist and focal position. Their mathematical definition are now given in the Methods section.

About the exact mechanism of g and γ modification: In the manuscript we have stated that: "If a curved pattern is set on the SLM, it imprints a curvature on the wavefront which, in turn, causes a weak lensing effect on the laser beam. The curvature is frequency dependent, and this way the size of the beam $W_0(\omega)$ on the final focusing parabola is changed. Consequently, g and γ are also affected". In the new version one can find a modified statement stressing out that this directly changes focal waists $w_0(\omega)$ and positions of foci $f(\omega)$.

Analytical expression, how g and γ change as the beam passes a thin lens are given in [8]. For example γ of a beam before and after a lens depends on the derivative of the distance of the lens from the collimated-beam waist with respect to the frequency as well as some other parameters. Therefore, we think that to obtain a good quantitative relationship between the applied SLM patterns, one would need to perform a complete characterization of the electric nearfield in space and frequency (amplitudes and phases), position of the focusing elements, and propagate the beam numerically to obtain relevant g and γ values. As such a spatiotemporal characterization of a beam is a highly complex task with unsecure results, we think that a detailed analysis is worth of a standalone study and we decided not to go to details in this paper.

List of bigger changes to the manuscript (highlighted with green font color in the manuscript text)

- Introduction last paragraph: mentioned that the samples differs from the Optica (2021) paper by electrode design and substrate.
- Paragraph 2.1: Pulse train CEP offset and measure phase equivalence.
- Paragraph 2.2: notion about the polarization state.
- New Figure 2: Part (c) of Fig.1 was moved in Fig. 2. Added panels describing the principle and robustness of the probe.
- Paragraph 3.1: w_0 and measured volume
- Paragraph 3.2: g and γ and their meaning
- Paragraph 4.2: g and γ change w_0 and f

References

1. M. A. Porrás, Z. L. Horváth, and B. Major, "Three-dimensional carrier-envelope-phase map of focused few-cycle pulsed Gaussian beams," Phys. Rev. A **98**, 063819 (2018).
2. A. Korobenko, S. Saha, A. T. K. Godfrey, M. Gertsvolf, A. Y. Naumov, D. M. Villeneuve, A. Boltasseva, V. M. Shalaev, and P. B. Corkum, "High-harmonic generation in metallic titanium nitride," Nat. Commun. **12**, 4981 (2021).
3. D. Hoff, M. Krüger, L. Maisenbacher, G. G. Paulus, P. Hommelhoff, and A. M. Sayler, "Using the focal phase to control attosecond processes," J. Opt. **19**, 124007 (2017).
4. D. Hoff, M. Krüger, L. Maisenbacher, A. M. Sayler, G. G. Paulus, and P. Hommelhoff, "Tracing the phase of focused broadband laser pulses," Nat. Phys. **13**, 947–951 (2017).
5. B. Alonso, M. Miranda, F. Silva, V. Pervak, J. Rauschenberger, J. San Román, Í. J. Sola, and H. Crespo, "Characterization of sub-two-cycle pulses from a hollow-core fiber compressor in the spatiotemporal and spatio-spectral domains," Appl. Phys. B Lasers Opt. **112**, 105–114 (2013).
6. V. Hanus, V. Csajbók, Z. Pápa, J. Budai, Z. Márton, G. Z. Kiss, P. Sándor, P. Paul, A. Szeghalmi, Z.

Wang, B. Bergues, M. F. Kling, G. Molnár, J. Volk, and P. Dombi, "Light-field-driven current control in solids with pJ-level laser pulses at 80 MHz repetition rate," *Optica* **8**, 570 (2021).

7. B. Major, M. A. Porras, and Z. L. Horváth, "Rotation of the polarization direction and reversal of helicity of ultrashort pulsed beams propagating in free space," *J. Opt. Soc. Am. A* **31**, 1200 (2014).
8. B. Major, Z. L. Horváth, and M. A. Porras, "Phase and group velocity of focused, pulsed Gaussian beams in the presence and absence of primary aberrations," *J. Opt.* **17**, 065612 (2015).

REVIEWERS' COMMENTS

Reviewer #1 (Remarks to the Author):

The authors have thoroughly responded to my (and, as far as I can see, the other referees') critique. The presentation has improved considerably in several paragraphs that had been not clear so far. Just one remark: "single-beam" in the meaning "non-interferometric" is used first in line 44, but non-interferometric only in line 47, without explaining the connection. I suggest to add in line 44 "(i.e. non-interferometric)" after the word "single-beam".

Reviewer #2 (Remarks to the Author):

In the opinion of this referee, the authors have corresponded adequately to the suggestions and criticism raised on their manuscript. The paper should now be ready for dissemination by Nature Communications.

Reviewer #3 (Remarks to the Author):

The authors carefully addressed the concerns I raised in my report and further improved their manuscript. I would like to recommend it for publication in Nature Communications now without further changes.

Reply to Reviewer 1

Comment:

The authors have thoroughly responded to my (and, as far as I can see, the other referees') critique. The presentation has improved considerably in several paragraphs that had been not clear so far. Just one remark: "single-beam" in the meaning "non-interferometric" is used first in line 44, but non-interferometric only in line 47, without explaining the connection. I suggest to add in line 44 "(i.e. non-interferometric)" after the word "single-beam".

Reply:

We added "non-interferometric" to the suggest place in the text.